# Interobserver Agreement of PD-L1/SP142 Immunohistochemistry and Tumor-Infiltrating Lymphocytes (TILs) in Distant Metastases of Triple-Negative Breast Cancer: A Proof-of-Concept Study. A Report on Behalf of the International Immuno-Oncology Biomarker Working Group

**DOI:** 10.3390/cancers13194910

**Published:** 2021-09-29

**Authors:** Mieke R. Van Bockstal, Maxine Cooks, Iris Nederlof, Mariël Brinkhuis, Annemiek Dutman, Monique Koopmans, Loes Kooreman, Bert van der Vegt, Leon Verhoog, Celine Vreuls, Pieter Westenend, Marleen Kok, Paul J. van Diest, Inne Nauwelaers, Nele Laudus, Carsten Denkert, David Rimm, Kalliopi P. Siziopikou, Scott Ely, Dimitrios Zardavas, Mustimbo Roberts, Giuseppe Floris, Johan Hartman, Balazs Acs, Dieter Peeters, John M.S. Bartlett, Els Dequeker, Roberto Salgado, Fabiola Giudici, Stefan Michiels, Hugo Horlings, Carolien H. M. van Deurzen

**Affiliations:** 1Department of Pathology, Cliniques Universitaires Saint-Luc, 1200 Brussels, Belgium; Mieke.vanbockstal@uclouvain.be; 2Department of Pathology, Erasmus Medical Center Cancer Institute, 3015 GD Rotterdam, The Netherlands; m.cooks@erasmusmc.nl; 3Division of Tumor Biology and Immunology, Netherlands Cancer Institute, 1066 CX Amsterdam, The Netherlands; i.nederlof@nki.nl (I.N.); m.kok@nki.nl (M.K.); 4Laboratory for Pathology East Netherlands, 7555 BB Hengelo, The Netherlands; m.brinkhuis@labpon.nl; 5Isala, 8025 AB Zwolle, The Netherlands; a.c.dutman@isala.nl; 6Pathology DNA, 5223 GZ ’s-Hertogenbosch, The Netherlands; m.koopmans@jbz.nl; 7Department of Pathology, Maastricht University Medical Center (MUMC), 6229 HX Maastricht, The Netherlands; loes.kooreman@mumc.nl; 8Department of Pathology, University Medical Center Groningen (UMCG), 9713 GZ Groningen, The Netherlands; b.van.der.vegt@umcg.nl; 9Reinier Haga Medical Diagnostic Center, 2625 AD Delft, The Netherlands; l.verhoog@reinier-mdc.nl; 10Department of Pathology, University Medical Center Utrecht (UMCU), 3584 CX Utrecht, The Netherlands; c.p.h.vreuls@umcutrecht.nl (C.V.); p.j.vandiest@umcutrecht.nl (P.J.v.D.); 11Pathology Laboratory, 3318 AL Dordrecht, The Netherlands; pwestenend@paldordrecht.nl; 12Department of Public Health and Primary Care, Biomedical Quality Assurance Research Unit, University of Leuven, Kapucijnenvoer 35d, 3000 Leuven, Belgium; inne.nauwelaers@kuleuven.be (I.N.); nele.laudus@kuleuven.be (N.L.); els.dequeker@kuleuven.be (E.D.); 13Institute of Pathology, Philipps-University Marburg and University Hospital Marburg (UKGM), Baldingerstr. 1, 35043 Marburg, Germany; carsten.denkert@uni-marburg.de; 14Department of Pathology, Yale School of Medicine, New Haven, CT 06510, USA; david.rimm@yale.edu; 15Section of Breast Pathology, Northwestern University, Chicago, IL 60611, USA; p-siziopikou@northwestern.edu; 16Translational Medicine, Bristol-Myers Squibb, Princeton, NJ 08540, USA; scott.ely@bms.com (S.E.); mustimbo.roberts@bms.com (M.R.); 17BMS Oncology Clinical Development, Bristol-Myers Squibb, Princeton, NJ 08540, USA; dimitrios.zardavas@bms.com; 18Department of Imaging and Pathology, Laboratory of Translational Cell & Tissue Research, KU Leuven–University of Leuven, 3000 Leuven, Belgium; giuseppe.floris@uzleuven.be; 19Department of Pathology, University Hospitals Leuven, 3000 Leuven, Belgium; 20Department of Oncology and Pathology, CCK, Karolinkska Institutet, 17177 Stockholm, Sweden; johan.hartman@ki.se (J.H.); balazs.acs@yale.edu (B.A.); 21Department of Clinical Pathology and Cytology, Karolinska University Laboratory, 17177 Stockholm, Sweden; 22HistoGenex NV, 2610 Antwerp, Belgium; dieter-peeters@telenet.be; 23Department of Pathology, AZ Sint-Maarten, 2800 Mechelen, Belgium; 24Ontario Institute for Cancer Research, Toronto, ON M5G OA3, Canada; john.bartlett@oicr.on.ca; 25Department of Laboratory Medicine and Pathobiology, University of Toronto, Toronto, ON M5S 1A8, Canada; 26Cancer Research UK Edinburgh Centre, Institute of Genetics and Cancer, The University of Edinburgh, Edinburgh EH4 2XR, UK; 27Department of Pathology, GZA-ZNA Hospitals, 2050 Antwerp, Belgium; roberto@salgado.be; 28Division of Research, Peter MacCallum Cancer Centre, Melbourne, VIC 8006, Australia; 29Department of Biostatistics and Epidemiology, Gustave Roussy, University Paris-Saclay, 94805 Villejuif, France; fgiudici@units.it (F.G.); stefan.michiels@gustaveroussy.fr (S.M.); 30Oncostat U1018, Inserm, University of Paris-Saclay, 94807 Villejuif, France; 31Division of Molecular Pathology, The Netherlands Cancer Institute, 1066 CX Amsterdam, The Netherlands; h.horlings@nki.nl; 32Department of Medicine, Yale School of Medicine, New Haven, CT 06510, USA

**Keywords:** PD-L1, SP142, triple-negative breast cancer, TNBC, tumor-infiltrating lymphocytes, TILs, immune cells, distant metastasis, atezolizumab, interobserver variability

## Abstract

**Simple Summary:**

Triple-negative breast cancer (TNBC) is an aggressive breast cancer subtype that lacks significant expression of estrogen receptor, progesterone receptor, and HER2. Patients with locally advanced or metastatic TNBC benefit from treatment with atezolizumab, a humanized monoclonal antibody that blocks the PD-L1 protein. Immunohistochemical analysis of the tumor microenvironment is essential to determine the amount of tumor-infiltrating PD-L1-positive immune cells. The PD-L1/SP142 clone is the companion diagnostic for atezolizumab. Here we investigate the degree of interobserver agreement among ten breast pathologists in the assessment of PD-L1/SP142 immunohistochemistry, as well as the assessment of tumor-infiltrating lymphocytes (TILs) in 49 metastatic TNBCs. This multicenter study shows that both PD-L1 assessment and TILs assessment are robust markers at the group level, but the observed interobserver variability likely affects treatment decisions for individual patients.

**Abstract:**

Patients with advanced triple-negative breast cancer (TNBC) benefit from treatment with atezolizumab, provided that the tumor contains ≥1% of PD-L1/SP142-positive immune cells. Numbers of tumor-infiltrating lymphocytes (TILs) vary strongly according to the anatomic localization of TNBC metastases. We investigated inter-pathologist agreement in the assessment of PD-L1/SP142 immunohistochemistry and TILs. Ten pathologists evaluated PD-L1/SP142 expression in a proficiency test comprising 28 primary TNBCs, as well as PD-L1/SP142 expression and levels of TILs in 49 distant TNBC metastases with various localizations. Interobserver agreement for PD-L1 status (positive vs. negative) was high in the proficiency test: the corresponding scores as percentages showed good agreement with the consensus diagnosis. In TNBC metastases, there was substantial variability in PD-L1 status at the individual patient level. For one in five patients, the chance of treatment was essentially random, with half of the pathologists designating them as positive and half negative. Assessment of PD-L1/SP142 and TILs as percentages in TNBC metastases showed poor and moderate agreement, respectively. Additional training for metastatic TNBC is required to enhance interobserver agreement. Such training, focusing on metastatic specimens, seems worthwhile, since the same pathologists obtained high percentages of concordance (ranging from 93% to 100%) on the PD-L1 status of primary TNBCs.

## 1. Introduction

Triple-negative breast cancer (TNBC) is an aggressive breast cancer subtype, characterized by the absence of estrogen receptor (ER) and progesterone receptor (PR) expression, as well as lack of gene amplification and/or protein overexpression of the human epidermal growth factor receptor 2 (HER2) [1]. This surrogate molecular subtype accounts for approximately 10–12% of diagnosed breast cancers [2,3,4,5]. Histologically, TNBCs are generally associated with a high tumor grade, a high proliferation index, and high levels of tumor-infiltrating lymphocytes (TILs) [1]. Compared with other breast cancer subtypes, metastatic TNBCs have limited treatment options and a poor outcome, as targeted therapies were lacking till recently [6]. Patients with metastatic TNBC generally have rapid progression with a median overall survival of only 13 months [7]. New treatment options such as PARP inhibitors and immunotherapy, i.e., immune checkpoint inhibitors targeting the PD-1/PD-L1 (Programmed Death-Ligand 1) axis, are now emerging [8,9,10,11]. PD-L1 is a transmembrane protein that downregulates anti-cancer immune responses through binding to PD-1, an inhibitory protein expressed by activated T-lymphocytes. The binding of PD-L1 to PD-1 prevents T-cell activation through the T-cell receptor [6]. In TNBC, phase 1/2 trials have shown response rates between 5–23% with anti-PD(L)1 monotherapy in heavily pretreated patients. In the TONIC-trial, a non-comparative phase 2 trial assessing nivolumab (anti-PD-1) monotherapy with or without induction treatment, the overall response rate in metastatic TNBC was 20% [12]. Importantly, the phase 3 IMpassion130 trial has shown that the addition of anti-PD-L1 (atezolizumab) to standard chemotherapy (nab-paclitaxel) in the first line of treatment for patients with advanced TNBC results in an overall survival benefit of seven months in patients with a PD-L1-positive tumor [13]. Recent biomarker analysis of IMpassion130 showed that TILs and PD-L1 status predict benefit to PD-L1 inhibition [13]. Similar observations were made in the KEYNOTE-355 trial, wherein pembrolizumab (anti-PD-1) treatment was associated with improved progression-free survival in patients with TNBCs with a combined positivity score of PD-L1/22C3 ≥10 [14]. These data illustrate how immune-oncology could change the treatment landscape for TNBC in the near future.

The VENTANA PD-L1/SP142 immunohistochemistry assay was developed to select patients who are likely to respond to atezolizumab and is approved by the US Food and Drug Administration (FDA) as a companion diagnostic assay [6,15]. At present, atezolizumab is added to nab-paclitaxel to treat patients with newly diagnosed PD-L1-positive locally advanced and/or metastatic TNBC [16], which indicates the need for optimal and reproducible measurement of PD-L1 expression. The use of the SP142 clone results in less tumor cell staining when compared with other PD-L1 clones such as SP263 and 22C3 [17], and its expression in breast cancer is therefore evaluated in immune cells only. Despite these observed differences, the SP263 and SP142 clones were shown to bind to an identical epitope in the cytoplasmic domain at the extreme C-terminus of the PD-L1 protein [18]. The prevalence of PD-L1/SP142-positivity in TNBC has recently been reviewed in detail by Gonzalez-Ericsson et al. and varies from 32% to 58% [19], except one phase 1 study with a positivity rate of 78% due to an initial selection bias [20]. The IMpassion130 trial has shown that PD-L1-positivity is lower in metastatic TNBC than in the primary tumor [13]. The PD-L1 status also strongly varies according to the localization of the metastasis, with the highest positivity rates reported for lymph node metastases [21].

PD-L1-positive TNBCs express PD-L1/SP142 in tumor-infiltrating immune cells covering ≥1% of the tumor area [15,21]. Considering that the PD-L1/SP142 status of breast cancers depends on the number of tumor-infiltrating immune cells, a combined TILs-PD-L1 assessment may be a logical path forward to pursue [19]. Previous studies of variable size have reported the interobserver reproducibility of PD-L1/SP142 assessment in breast cancer samples, and their results varied from poor to very good [22,23]. Interobserver variability seems more limited within a single institution [23]. These discrepant results could be explained by the heterogeneous set-up of these studies, with respect to the training and the number of the participating pathologists, and the number of TNBC samples evaluated. The degree of agreement also depends on how the concordance evaluation was performed, as the use of tissue micro-arrays vs. whole tissue slides, and analysis of categorical vs. continuous variables may affect concordance results. Industry-sponsored studies could be biased towards the interests of the sponsor of the study [24]. The conclusions of biomarker research performed on materials from clinical trials must be interpreted irrespective of the companion diagnostic that was FDA-approved for that particular study. It can be debated whether a biomarker that was highly concordant after categorical analysis using a very low cut-off (of 1%, for instance), and then poorly concordant when analyzed as a continuous variable, should be considered as an analytically valid biomarker, even if the clinical application is based on that particular cut-off. Most importantly, a close examination of most biomarkers suggests a continuous relationship between the likelihood of response to therapy and the level of biomarker expression. That is, even within the group of “responders”, patients with higher levels of biomarker expression are more likely to benefit from treatment than those with a lower but still “positive” result. This suggests an urgent need to develop standards for powered and unbiased concordance analysis across quantitative biomarkers in pathology. As clinicians continue to use, almost universally suboptimal, cut-points for binary “treatment/no treatment” decisions, this emphasizes the need to develop clinically robust assay quantitation that can subsequently be robustly tested for concordance by the scientific community in a partnership with industry and the regulatory authorities to allow the appropriate treatment decision to be informed by informed analysis of risk vs. benefit by individual patients and their clinicians [25]. However, just because a treatment decision is binary (treatment/no treatment), it does not necessarily mean we need to mindlessly pursue binary cut-points for biomarkers. In fact, for most biomarkers, the probability of a patient group responding increases with the increasing positivity of the biomarker, e.g., patients with strongly ER-positive cancers are more likely to respond to endocrine therapy than weakly ER-positive cancers, and this applies to many biomarkers, including TILs and PDL1. In a computerized age, we may potentially do better than lumping all patients with a “positive” biomarker based on a randomly selected binary cut-point into the “responders” group and everyone else into the “non-responders” group.

Previous PD-L1 concordance studies were mainly performed on the primary tumor, while the PD-L1/SP142 assay is often performed on biopsies from distant metastases, both in clinical trials and in daily routine practice. In IMpassion130, PD-L1 was assessed in the primary tumor in 62% of patients, whereas in 38% of the patients, PD-L1 was assessed in different metastatic sites, accessed from lymph nodes, lung, liver, soft tissues, and skin [26]. Distant metastases are heterogeneous with respect to tissue composition, immune landscape, the amount of stroma, and tumor quantity. Previous studies reported that distant metastases have lower numbers of TILs compared with the primary tumor [13,27], and TILs levels also differ according to the anatomical location [21]. The tissue type selected for the PD-L1/SP142 assay may therefore influence the subsequent treatment decision. Since PD-L1/SP142 expression is evaluated in tumor-infiltrating immune cells, there is an association between the number of TILs and the PD-L1 positivity status [6]. Metastatic TNBC without TILs is less likely to present with PD-L1 expression. As such, the combined analysis of TILs and PD-L1 expression in TNBC could have added value for the selection of TNBC patients for PD-L1 targeted therapy. In this national multicenter study, we investigated the inter-pathologist agreement of TILs assessment and PD-L1/SP142 scoring in distant metastases of TNBC.

## 2. Materials and Methods

### 2.1. PD-L1/SP142 Proficiency Test

The PD-L1/SP142 proficiency test comprised an evaluation of 28 digitalized PD-L1/SP142-stained slides. The proficiency test was preceded by training: both were organized in The Netherlands in 2019 by the manufacturer of the PD-L1/SP142 immunohistochemical assay (Roche Diagnostics GmbH, Mannheim, Germany). The proficiency set comprised two biopsy specimens and 26 resection specimens. An ad hoc consensus diagnosis was available for each case. All digitalized slides were present on a password-protected online platform (slidescore.com, Version 1.1, Slide Score B.V., Amsterdam, The Netherlands). All ten participating Dutch breast pathologists passed the proficiency test (wherein a maximum of two discordant results were allowed), which was a condition to participate in the present study.

### 2.2. Specimens in the Study Set & Slide Scanning

Formalin-fixed, paraffin-embedded (FFPE) tissue samples were collected from the archives of the departments of pathology of three different academic institutions: 23 samples from the Erasmus Medical Center Cancer Institute (Rotterdam, The Netherlands), 9 samples from the Netherlands Cancer Institute (Amsterdam, The Netherlands), and 17 samples from the University Medical Center Utrecht (Utrecht, The Netherlands). Tissue specimens comprised either breast cancer metastases with a known triple-negative immunohistochemical profile, or breast cancer metastases from patients with a history of TNBC. Both biopsies and resection specimens were eligible. Bone metastases were excluded if the tissue samples had undergone decalcification. Sample selection was based on the availability of leftover material, with enrichment for PD-L1 positive metastases to ensure a balanced study set of positive and negative cases. Patients had been diagnosed with metastatic TNBC between 1 January 1990 and 31 December 2020. Triple-negative breast carcinoma was defined as breast cancer with <10% tumor nuclei showing immunoreactivity for ER and PR, according to the Dutch Breast Cancer Guideline [28], and HER2-negativity as defined by the ASCO/CAP guidelines [29]. The use of this anonymized leftover patient material was in accordance with the Code of Conduct of the Federation of Medical Scientific Societies in the Netherlands (FEDERA) [30], as previously described [31]. The study methodologies were in line with the standards set by the Declaration of Helsinki.

Digitalization of hematoxylin & eosin-stained (H&E) tissue slides and PD-L1/SP142-stained tissue slides was performed by whole slide scanning using the Aperio slide scanner (Leica Biosystems, Buffalo Grove, IL, USA) whereafter the slides were uploaded to Slide Score (slidescore.com, Version 1.1, Slide Score B.V., Amsterdam, The Netherlands).

### 2.3. TILs Assessment

The extent of the stromal inflammatory infiltrate in the digitalized H&E slide of each metastasis was assessed according to the standardized method for TILs in breast cancer as described in detail by the International Immuno-Oncology Biomarkers Working Group [32,33]. Briefly, the density of stromal TILs (sTILs) was evaluated as the overall percentage (range 0–100%) of the stromal area within the borders of the tumor that is occupied by mononuclear immune cells. The total peri- and intra-tumor stromal surface area served as a denominator [32]. The number of fields for evaluation was not specified: all participants had to evaluate the entire area occupied by invasive carcinoma. In the case of heterogeneity, an average TILs score was noted. All participants evaluated the same set of digital slides. TILs were only evaluated in the study set, and not in the proficiency set.

### 2.4. PD-L1/SP142 Immunohistochemistry

Immunohistochemistry was centrally performed on 4-µm thick whole slide sections from FFPE tissue blocks, on a validated and accredited automated slide stainer (Benchmark ULTRA System, VENTANA Medical Systems, Tucson, AZ, USA) according to the manufacturer’s instructions. Briefly, following deparaffinization and heat-induced antigen retrieval, the tissue samples were incubated with the monoclonal rabbit anti-PD-L1 antibody (clone SP142, VENTANA Medical Systems) for 32 min at 37 °C, followed by hematoxylin II counter stain for 12 min and a blue coloring reagent for 8 min. Visualization was obtained by the OptiView DAB IHC Detection Kit (VENTANA Medical Systems). Each tissue slide contained a fragment of FFPE tonsil as an on-slide positive control. The assessment of PD-L1/SP142 immunohistochemistry was conducted according to the manufacturer’s instructions as mentioned in the package insert for the PD-L1/SP142 assay. These instructions comprised an estimation of the immune cells (IC) with discernable staining for PD-L1/SP142 (regardless of the intensity) covering <1% (negative) or ≥1% (positive) of the tumor area occupied by tumor cells, associated intra-tumoral and contiguous peritumoral stroma. The following formula was applied: ICarea(%) = (area occupied by positive IC of any type)/(total tumor area) × 100. These scoring criteria were applied in IMpassion130 [13].

### 2.5. Statistical Analysis

All sTILs and PD-L1/SP142 scores were collected in Excel (Excel Windows 10, Microsoft Corporation, Redmond, WA, USA) and imported into the IBM SPSS Statistics 26.0 software (IBM Chicago, IL, USA) for statistical analysis. Tests for normality were performed with the Shapiro–Wilk test, which showed that both the sTILs scores and PD-L1/SP142 scores of each participant were not normally distributed (*p* < 0.05) in either the proficiency set or the study set. Because of this non-normal distribution, the median (instead of the mean) values were selected for each case to serve as the “gold standard” in the study set, as a consensus diagnosis was lacking for this patient cohort. This hypothetical “median/gold standard” pathologist was designated “Px”. Descriptive measures, comprising minimum, maximum, and percentiles, were calculated.

Interobserver agreement was quantified by calculation of the intraclass correlation coefficients (ICC) for sTILs scores in the study set, and PD-L1/SP142 scores in both the proficiency set and the study set. Interpretation of the ICC was performed in line with the proposal of Koo and Li [34]. ICC settings were: two-way random, single measures, absolute agreement. The overall ICC value and corresponding 95% confidence interval (CI) were also determined for subgroups according to the localization of metastases. Bland-Altman plots were constructed to visualize the degree of deviation from the median score Px for both sTILs and PD-L1/SP142 [35], by using both the mean of and the difference between each pathologist’s score and the median Px score (study set) or consensus score (proficiency set). Percent agreement based on PD-L1 status (i.e., positive vs. negative) was calculated and compared with the consensus diagnosis (proficiency set) or median Px diagnosis (study set). Cohen’s Kappa values were calculated per pathologist duo for PD-L1 status in both the proficiency set and the test set, and interpretation was performed according to Landis and Koch [36]. Concordance rates for each pair of pathologists were calculated at several TILs (%) cut-offs: <5% vs. ≥5%, <10% vs. ≥10%, <30% vs. ≥30% and <75% vs. ≥75%. Computations were not performed for the 1% thresholds as there was an unbalanced proportion between positive and negative classification; only 12 values out of 490 were assessed as <1%. For PD-L1 assessment scores, concordance rates for each pair of pathologists were calculated at three thresholds: <1% vs. ≥1%, <5% vs. ≥5%, and <10% vs. ≥10%. Each concordance rate was the percent agreement derived from a 2 × 2 contingency table created for each cut-off and pair of pathologists. The results were reported as the sample mean and sample standard deviation of these concordance rates for all pairs of pathologists. A custom code has been created for the concordance analysis using the R software environment for statistical computing and graphics (version 4.0.2, R Foundation, Vienna, Austria).

The Mann–Whitney U test was used to explore the association between TILs scores and PD-L1 status. The Kruskal–Wallis test was used to explore the association between TILs scores and the localization of the metastases. Visualization was performed by the construction of Box-and-Whisker plots. The Spearman’s Rho correlation test was used to investigate the relation between the median TILs score and the median PD-L1 score. A corresponding scatter plot was constructed. All tests were two-sided and significance levels were set at α = 0.05, except for the Kruskal–Wallis test, where we applied a post hoc Bonferroni correction for multiple group testing (α = 0.0083).

## 3. Results

### 3.1. Composition of the Proficiency Set and Study Set

The proficiency set contained 28 samples in total, including two breast biopsies (7%) and 26 breast resection specimens (93%). All breast cancers belonged to the triple-negative subtype as assessed according to the Dutch national breast cancer guidelines. As determined by consensus, the proficiency set contained 16 PD-L1-positive cases (57%) and 12 PD-L1-negative cases (42%), with PD-L1 scores ranging from 0% to 20%. The median score amounted to 3.0%.

The study set comprised 49 breast cancer metastases, of which 40 (82%) had a known triple-negative status. The remaining 9 (18%) breast cancer metastases had developed in patients with a history of TNBC. The study set comprised 17 (35%) biopsies, 20 (41%) resection specimens, and 12 (25%) tissue samples where the technique had not been specified. Breast metastases originated from the brain (12 cases; 25%), the liver (2 cases; 4%), the lungs (7 cases; 12%), lymph nodes (9 cases; 18%), the skin (11 cases; 22%), soft tissues (4 cases; 8%), the sternum (one case; 2%), the kidney (one case; 2%), the colon (one case; 2%) and the terminal ileum (one case; 2%). As a consensus PD-L1 score was lacking for this set, the median PD-L1 score Px was selected as a surrogate ‘gold standard’ in the study set, based on the assessment of ten pathologists. This resulted in 20 (41%) PD-L1-positive and 29 (59%) PD-L1-negative TNBC metastases. The median PD-L1 score Px, therefore, corresponded with 0%. Both the consensus scores in the proficiency set and all pathologists’ PD-L1 scores in the proficiency set and study set were not normally distributed, as determined by the Shapiro-Wilk test (*p* < 0.05).

### 3.2. Interobserver Agreement for PD-L1 Assessment in the Proficiency Set

All ten participating pathologists evaluated all 28 TNBC cases in the proficiency test. There were no missing values. The overall ICC was 0.510 (95%CI 0.370–0.673), indicating borderline moderate agreement for PD-L1/SP142 assessed as percentages. Figure 1 shows the distribution of the PD-L1/SP142 percentages per TNBC in the proficiency set. PD-L1-negative TNBCs (Figure 2) generally presented with a range equaling zero, but the range between the 25th and 75th percentiles (P25 and P75, respectively) were substantially larger for PD-L1-positive TNBCs, which resulted in increased interobserver variability.

The median PD-L1/SP142 score in the proficiency set, based on the evaluation of all ten participants, correlated significantly with the consensus diagnosis (Spearman’s Rho = 0.956; *p* < 0.001; Figure 3). The ICC values for the PD-L1 score of each pathologist in comparison with all the other pathologists, as well as the consensus score and the median Px PD-L1 score are shown in Table 1. The PD-L1 assessment of nine out of ten (90%) pathologists showed good agreement when compared with the consensus score, and agreement was moderate for one pathologist (10%). The corresponding Bland-Altman plots visualize the deviations from the consensus score (Appendix A). Similarly, the PD-L1 assessment of seven (70%) pathologists showed good agreement with the median Px PD-L1 score, whereas agreement was moderate for one pathologist (10%) and excellent for two pathologists (20%).

Mutual agreement among these ten pathologists ranged from poor to excellent, with the lowest ICC value equaling 0.386 and the highest ICC value equaling 0.922. However, dichotomization of the PD-L1 scores as percentages into a positive vs. negative PD-L1 status resulted in substantial percentage agreement with the consensus score. The PD-L1 assessment of five (50%) pathologists was 100% concordant with the consensus score. Two (20%) and three (30%) pathologists showed 93% and 96% agreement with the consensus score, respectively. Of note, all ‘deviations’ from the consensus score in the proficiency set were due to overestimation of the percentage of PD-L1-positive IC. In other words, all ‘wrong’ answers were due to false-positive PD-L1 assessments. The high percentage agreement with the consensus score resulted in high corresponding Cohen’s Kappa values (Ƙ ≥ 0.851).

### 3.3. Interobserver Agreement for PD-L1 Assessment in the Study Set

The overall ICC value for PD-L1 assessment as a percentage amounted to 0.299 (95%CI 0.197–0.430), indicating poor interobserver agreement. Subsequently, breast cancer metastases were reclassified in four different groups: skin and soft tissue metastases, brain metastases, lymph node metastases, and ‘other’ localizations. ICC values for each subgroup were 0.172, 0.144, 0.364 and 0.313, respectively. Although the ICC value for lymph node metastases was slightly higher than the ICC values of other metastatic localizations, there was still poor agreement for PD-L1 assessment as a percentage. Descriptive values of the assessment are shown in Appendix A. The variable interobserver agreement was reflected in the ICC values per pathologist duo (Table 2), which ranged from 0.167 (poor agreement) to 0.944 (excellent agreement). Corresponding Bland-Altman plots can be found in Appendix A.

However, an agreement was generally higher when the assessment of each individual pathologist was compared with the median PD-L1/SP142 score, which was used as a ‘surrogate’ consensus score in the study set. Four (40%) pathologists showed excellent agreement with the median Px PD-L1 score, whereas five (50%) and one (10%) pathologists showed good and moderate agreement, respectively (Table 2). The dichotomization of the percentage scores into a positive vs. negative PD-L1 status resulted in a relatively high percent agreement with the median Px PD-L1 score, ranging from 80% to 94%. Deviations of the median score were due to both ‘false-positive’ and ‘false-negative’ scores. Mutual comparison of all pathologists showed variable Cohen’s Kappa values, ranging from 0.383 to 0.952 (Table 3). There was 100% agreement on the PD-L1 status of 23 metastatic TNBCs (47%). Agreement on the PD-L1 status was 90% and 80% for nine (18%) and four (8%) cases, respectively. Three (6%) and nine (18%) metastatic TNBCs showed an agreement on the PD-L1 status of 70% and 60%, respectively. One case (2%) was considered PD-L1-negative by five pathologists, and PD-L1 positive by the five other pathologists, resulting in an agreement of 50%. The 10 metastatic TNBCs with at least 40% discordance originated from the skin (4 cases), brain (4 cases), liver (1 case), or a lymph node (1 case).

### 3.4. Interobserver Agreement for TILs Assment in the Study Set

The TILs levels in the study set ranged from 0% to 99%, as assessed by all ten participating pathologists. Descriptive values are provided in Appendix A. The overall ICC value for TILs was 0.517 (95%CI 0.408–0.639), which indicates moderate agreement. Subsequently, an agreement was investigated among breast cancer metastases per localization: skin and soft tissue metastases, brain metastases, lymph node metastases, and ‘other’ localizations. ICC values for each subgroup were 0.530, 0.489, 0.463, and 0.523, respectively. TILs assessment seems slightly more difficult in brain and lymph node metastases as these subgroups showed only poor agreement. The ICC values per pathologist duo are provided in Table 4. Because of the lack of an ad hoc consensus TILs score, the median TILs score Px is used as a surrogate consensus score. Importantly, the TILs scores of each individual pathologist show good agreement with the surrogate consensus score Px in nine out of ten (90%) participants, and excellent agreement for one (10%) pathologist.

TILs were not determined in the proficiency set, as several previous studies have already reported on the interobserver variability in TILs assessment in primary TNBC [37,38,39].

### 3.5. Exploratory Analysis of Different Thresholds for TILs and PD-L1 Assessment

The impact on interobserver agreement was explored for four different thresholds for dichotomization of TILs levels in the study set (Table 5). The 5% and 10% threshold resulted in overall moderate concordance among the ten participants, whereas agreement was good for the 30% threshold. The 75% cut-off for dichotomized TILs scores resulted in excellent interobserver concordance.

Similarly, three different thresholds for dichotomization of PD-L1 scores were investigated in both the proficiency set and the study set (Table 6). The 1% cut-off resulted in excellent interobserver concordance in the proficiency set, and good concordance in the study set.

### 3.6. Associations between Metastatic Location, PD-L1 and TILs Assessment in the Study Set

The median TILs score Px was significantly associated with the median Px PD-L1 status (*p* = 0.004). TNBC metastases with negative PD-L1 status present substantially fewer TILs than TNBCs with positive PD-L1 status, although there is a discrete overlap in the TILs levels between both groups (Figure 4). Although not statistically significant (*p* = 0.136), we observed a trend towards higher TILs levels in lymph node metastases than in brain metastases or skin and soft tissue metastases. Skin and soft tissue metastases generally presented with the lowest TILs levels (Figure 5). The heterogeneous group of ‘other’ metastatic localizations showed a wide range of TILs levels, but liver, bone, lung, kidney, and bowel metastases were each represented in too small numbers to allow for reliable statistical analysis.

Brain metastases had significantly lower PD-L1 scores than lymph node metastases (*p* < 0.001) or skin and soft tissue metastases (*p* = 0.001). Lymph node metastases did not show significantly different PD-L1 scores than skin and soft tissue metastases (*p* = 0.010 after Bonferroni correction), although there was a discrete trend towards higher PD-L1 scores in the former. Lymph node metastases did not show significantly different PD-L1 scores from TNBC metastases in ‘other’ locations (*p* = 0.841). Skin and soft tissue metastases, as well as brain metastases, showed significantly lower PD-L1 percentages than TNBC metastases in ‘other’ locations (*p* = 0.008 and *p* < 0.001, respectively).

Comparison of the degree of interobserver agreement for PD-L1 assessment between the proficiency set and the study set per pathologist shows that the overall agreement seems similar for eight out of ten (80%) participants (Figure 6). The PD-L1 scores of P2 showed a higher agreement with the median PD-L1 scores in the study set than in the proficiency set, whereas the opposite was true for P5.

### 3.7. Combined Evaluation of Dichotomous PD-L1 and TILs Assessment in the Study Set

As a final exploratory analysis, we created a combined biomarker, based on the PD-L1/SP142 status (positive vs. negative using the <1% vs. ≥1% cut point) and dichotomized TILs assessment (low vs. high TILs using an arbitrary <5% vs. ≥5% cut point). Only those TNBC metastases with ≥5% TILs and a positive PD-L1 status were considered positive for this combined biomarker. This analysis is merely a preliminary proof-of-concept evaluation, as there is no clinical evidence available at present in favor of a particular TILs cut point. The combined biomarker did not reduce the degree of inter-observer discordance at the individual patient level. The frequency of a ‘positive’ combined biomarker varied from 11 out of 49 (22%) cases to 23 out of 49 (47%) cases among the participants. All pathologists agreed on the combined biomarker status for 22 out of 49 metastases (45%). One and two pathologists disagreed with the other participants for ten (20%) and two (4%) out of 49 metastases, respectively. Three and four pathologists disagreed with the other participants for eight (16%) and six (12%) out of 49 cases, respectively. In one metastasis, five participants considered the combined biomarker as positive, whereas the other five pathologists considered it as negative. This corresponded to an average kappa value of 0.568 (range 0.355–0.852; Table 7). The limited size of the study cohort precluded any further detailed analysis.

## 4. Discussion

In the present study, we investigated the inter-pathologist agreement for both TILs assessment and evaluation of PD-L1/SP142 immunohistochemistry. We focused on lymph node metastases and distant metastases from several anatomic localizations, as the PD-L1/SP142 immunohistochemical assay is often performed on metastatic TNBC in daily routine practice. We show here that TILs assessment in metastatic TNBC, performed by the method as proposed by the International Immuno-Oncology Biomarker Working Group, results in only moderate agreement among ten Dutch breast pathologists, even though all passed a PD-L1/SP142 proficiency test. The degree of interobserver variability seems similar to the one observed in TILs evaluation in primary TNBCs [40]. Although earlier reports questioned the influence of this interobserver variability on the predictive value of TILs for achieving a pathological complete response (pCR) in the neoadjuvant setting [37,39], it was recently shown that TILs assessment according to the Working Group’s method is a robust predictive marker for pCR at the group level [41]. The predictive value of TILs for pCR is not negatively affected by inter-pathologist disagreement [38], but other patient-related characteristics such as high body mass index may have an adverse impact on the prediction of pCR and remain to be further elucidated [42]. We, therefore, deem it unlikely that the presently observed degree of interobserver variability in TILs assessment would negatively influence the treatment response in metastatic TNBC at the group level. Nevertheless, this hypothesis will require confirmation in future studies, with emphasis on the consequences of treatment decisions for individual patients.

We observed a higher degree of interobserver variability in the assessment of PD-L1/SP142 immunohistochemistry in TNBC metastases among the ten participating pathologists, resulting in poor mutual concordance, which is in line with previously reported findings in a series of primary TNBC [22]. However, a recent study on PD-L1 assessment in primary TNBC reported a surprisingly high interobserver and intra-observer agreement, which is likely due to the use of tissue microarrays instead of whole slide immunohistochemistry [43]. Although the agreement for PD-L1/SP142 scores as percentages is poor, regardless of the site of the metastasis, the overall agreement on positive vs. negative PD-L1 status seems slightly better. The concordance of each individual pathologist’s scores with the ad hoc consensus diagnosis for primary TNBCs in the proficiency set, as well as with the median PD-L1/SP142 scores for metastatic TNBCs in the study set, was good to excellent at the group level, even when taking into account a more stringent interpretation of the ICC values [44]. Of note, all ‘deviations’ from the consensus score in the proficiency set were due to overestimation of the percentage of PD-L1-positive IC. This implies that one to two patients in the proficiency set would have been ‘overtreated’ due to so-called false-positive results, but none of the patients would have been denied treatment as there were no false-negative results. At first sight, the observed interobserver variability seems therefore acceptable at the group level.

We did however observe a negative impact of the interobserver variability on the PD-L1-positivity status at the level of the individual patient in the study set with metastatic TNBCs. Perfect agreement among the ten observers was noted for only 47% of all TNBC metastases, whereas one to five observers provided a discordant PD-L1-positivity status for the remaining 53% of this patient series. This observation indicates that dichotomization of the PD-L1 percentage score does not resolve the impact of the discordance observed among the scores as a percentage. Consequently, a substantial number of patients would not receive atezolizumab while they are likely to benefit, or vice versa. As already identified in primary TNBC by Reisenbichler et al., several patients would be assigned to the incorrect PD-L1 status depending on which pathologist read the immunohistochemical PD-L1/SP142 assay [22]. This phenomenon is inherent to each dichotomously assessed biomarker, and, though inevitable, it should be questioned which rate of discordance is clinically acceptable. The issue is not whether the obvious cases can be diagnosed, but whether the difficult cases can be diagnosed. At present, the answer seems to be “no”. In our series, four to five pathologists disagreed with the other observers regarding the PD-L1 status in ten out of 49 metastatic TNBC (20%). These challenging ‘borderline’ cases often originated from the skin or the brain. The difficulties in PD-L1 assessment encountered in these metastases could be explained by the generally low level of TILs in ‘immune-deserted’ TNBC metastases located in the skin, soft tissues, and brain, in comparison with the higher TILs levels in lymph node metastases. Skin and brain metastases are likely more often problematic for robust PD-L1 assessment, as fewer TILs are associated with fewer PD-L1-positive immune cells: these metastases often flirt with the 1% threshold for PD-L1 positivity. Our findings confirm previously reported observations on the distribution of PD-L1-positive immune cells in various metastatic sites [13,21]. TNBC metastases harbor additional difficulties, i.e., the intra- and peri-tumor stroma have a different aspect than the stroma in primary tumors in the breast, and its quantity can be very limited. Lymph node metastases represent a particular challenge, as lymph nodes are per definition characterized by high levels of lymphocytes in comparison with primary TNBCs [33].

The study set contained 41% PD-L1 positive and 59% PD-L1 negative metastases, resulting in a median PD-L1 score of 0%. This is not surprising, considering that most metastases contain few TILs, and consequently, contain very few PD-L1-positive immune cells. Our data suggest that both a sufficient number of positive vs. negative tissue samples and a sufficient number of pathologists are required for definitive concordance assessment, to allow for a powered analysis. The present proof-of-concept study includes only a limited number of tissue samples and pathologists; we, therefore, aim to help design future robust concordance studies in the metastatic setting. Although we did not assess specific pitfalls in the present study, TILs assessment and PD-L1 immunohistochemical evaluation are likely prone to similar pitfalls as described for primary TNBC [40]. Additional training for challenging metastatic TNBCs might enhance interobserver agreement. Kirkegaard et al. have indeed shown that with appropriate training, even with quantitative assessments, really high concordance can be achieved [44]. If we fail to address the importance of training, we will ultimately pave the way for automated image analysis to become the norm- simply because we failed to address the challenges posed by manual assessment without robust standards.

Such training, focusing on metastatic specimens, seems worthwhile, as the same observers achieved high percentages of agreement (ranging from 93% to 100%) on the PD-L1 status of a set of 28 primary TNBCs, as demonstrated in the proficiency set. This could indicate specific challenges associated with differences in the intra- and peri-tumor stroma between primary TNBCs and TNBC metastases. Other well-known ‘immune-deserted’ metastatic localizations, such as the liver, should be included as well. Specific guidelines might be required for bone metastases, including optimal specimen handling for PD-L1 assessment.

As TNBCs with high TILs are more likely to contain ≥1% PD-L1-positive immune cells [45], a combined evaluation of TILs and PD-L1 expression in locally advanced and metastatic TNBC has been proposed by the International Immuno-Oncology Biomarker Working Group to reduce the risk of suboptimal patient selection for immunotherapy [19]. Suboptimal patient selection might result from inter-pathologist variability in the PD-L1/SP142 assessment. The limited number of metastatic TNBCs included in the present study set precluded any robust analysis of combined evaluation of TILs and PD-L1 expression. Nevertheless, future studies should explore whether integrated training for both TILs and PD-L1 immunohistochemistry could enhance interobserver concordance, and its impact on PD-L1 immunohistochemistry-based treatment decisions. Similar to pending computational quantitative TILs assessment, the evaluation of PD-L1/SP142 immunohistochemistry might be objectified by future application of automated algorithms or so-called ‘artificial intelligence’ [46]. TNBC is not the only type of tumor posing challenges for robust and reliable PD-L1/SP142 assessment. Similar degrees of inter-pathologist discordance have been reported for carcinomas in other organs, including pulmonary adenocarcinomas, squamous carcinomas of the head and neck region, urothelial carcinomas, hepatocellular carcinomas, and gastric carcinomas [47,48,49,50,51,52]. Additionally, treatment response to atezolizumab does not only depend on PD-L1-positive immune cells. Future research should investigate the added value of other biomarkers too. For instance, Bocanegra et al. have demonstrated substantial differences in the percentage of PD-L1-positive CD11-positive myeloid cells between non-small cell lung cancer patients responding or not responding to immunotherapy [53]. Similar analyses should be performed in TNBC.

This proof-of-concept study was designed to inform pathologists on subsequent powered concordance studies aiming to provide more definitive conclusions on the concordance and use of PD-L1 and TILs in the metastatic setting. The observed concordances in this study are therefore merely indicative in nature, and all these available data suggest, as the most important message, that systematic training of pathologists is urgently required to enhance the degree of interobserver agreement. Additionally, adequate concordance analyses require sufficient knowledge of appropriate statistical methods to analyze and interpret the data in a correct way. A clinically useful analytical validity takes into account the interobserver agreement, as well as sensitivity, specificity, and negative and positive predictive value as determined by state-of-the-art methods. Integration of this information will be a major challenge for future validity studies for promising biomarkers such as TILs and PD-L1.

## 5. Conclusions

In this study, we observed an acceptable degree of inter-pathologist agreement for TILs assessment as percentages in TNBC metastases, but we detected substantial inter-pathologist variability in the assessment of PD-L1/SP142 immunohistochemistry as a percentage, as well as the PD-L1/SP142 status (positive vs. negative). It needs to be acknowledged that there is still no clear-cut definition of what constitutes a “good” or “acceptable” concordance between pathologists, and whether the clinical implications of the discordances observed should be factored in or not. From a purely analytical point of view, the clinical implications may not be considered, but in the clinical daily practice setting, it can be argued that the clinical use of the biomarker should drive this definition.

As previously reported, the TILs level was associated with the localization of the metastases. TNBC metastases with higher TILs were more likely to contain PD-L1/SP142-positive immune cells. The degree of interobserver variability in the assessment of PD-L1/SP142 immunohistochemistry as a percentage did not substantially affect the PD-L1 status at the group level. However, there was a considerable impact on the PD-L1 status (positive vs. negative) for individual patients, wherein the potential subsequent treatment decision would be dependent on the pathologist reading the biopsy. Despite having completed a prior training, ten pathologists could not agree on the treatment choice for over half of patients, and for one in five patients, the chance of treatment or not (positive vs. negative PD-L1) was essentially random, since half of the participants called them positive and half negative. This represents a significant diagnostic issue that requires urgent attention and implies the need for additional training with an emphasis on the assessment of TILs in metastatic TNBCs.

Training for integrated assessment of both TILs and PD-L1 immunohistochemistry should be investigated as it is biologically unlikely that TILs-negative TNBC metastases present with PD-L1-positivity. An integrated analysis might therefore increase the interobserver agreement. It seems worthwhile to explore future developments in digital pathology as computational assessment of both TILs and PD-L1/SP142 immunohistochemistry could help in optimizing the inter-pathologist agreement or provide a robust and portable alternative to pathologist-based interpretation. Finally, future proficiency tests, as well as External Quality Assessment (EQA) schemes, on TILs and PD-L1 need to ensure that the lessons learned from this study are taken into consideration; in particular the differences in concordance when analyzing a biomarker as a continuous vs. a categorical variable, and how the biomarker is used for clinical decision-making.

## Figures and Tables

**Figure 1 cancers-13-04910-f001:**
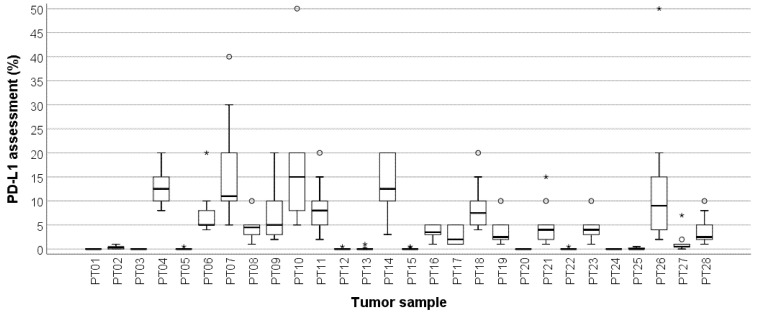
Box-and-whisker plot, illustrating the PD-L1/SP142 scores of ten pathologists per proficiency test (PT) case. The range between the 25th and 75th percentiles is 0 for PD-L1-negative TNBCs and increases substantially for PD-L1-positive cases. Circles represent outliers; asterisks represent extremes. The thick line within each box is the 50th percentile (i.e., median, * outliers, ° extremes).

**Figure 2 cancers-13-04910-f002:**
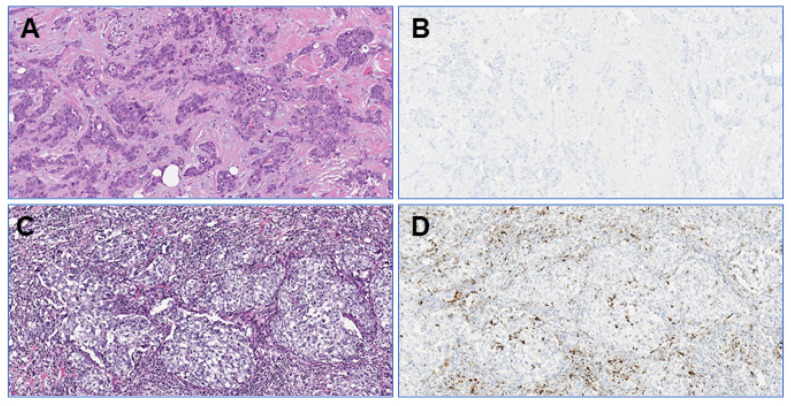
Photomicrographs (original magnification: 100×) of a TNBC in the proficiency set with low TILs (**A**) and no PD-L1/SP142-positive immune cells (**B**). The stroma of another TNBC shows high TILs levels (**C**) with several PD-L1/SP142-positive immune cells (**D**).

**Figure 3 cancers-13-04910-f003:**
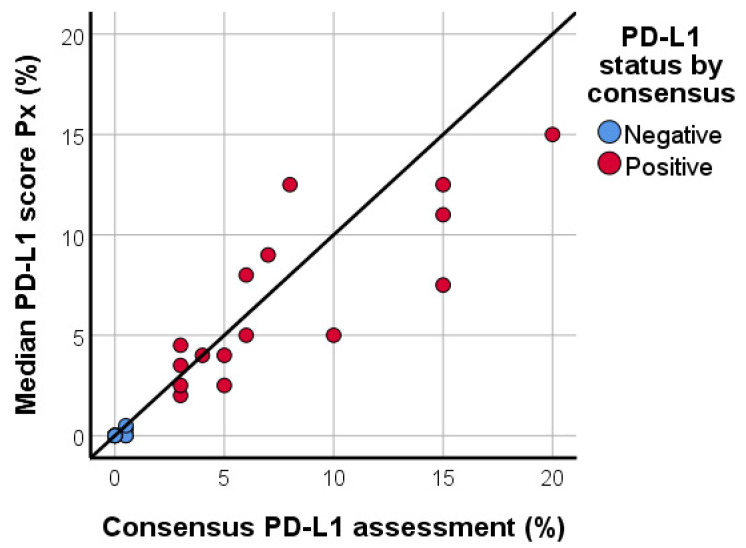
Scatter plot, illustrating the median PD-L1/SP142 score based upon the assessment by ten pathologists in relation to the consensus PD-L1 score in the proficiency test. Correlation is high (Spearman’s Rho = 0.956; *p* < 0.001).

**Figure 4 cancers-13-04910-f004:**
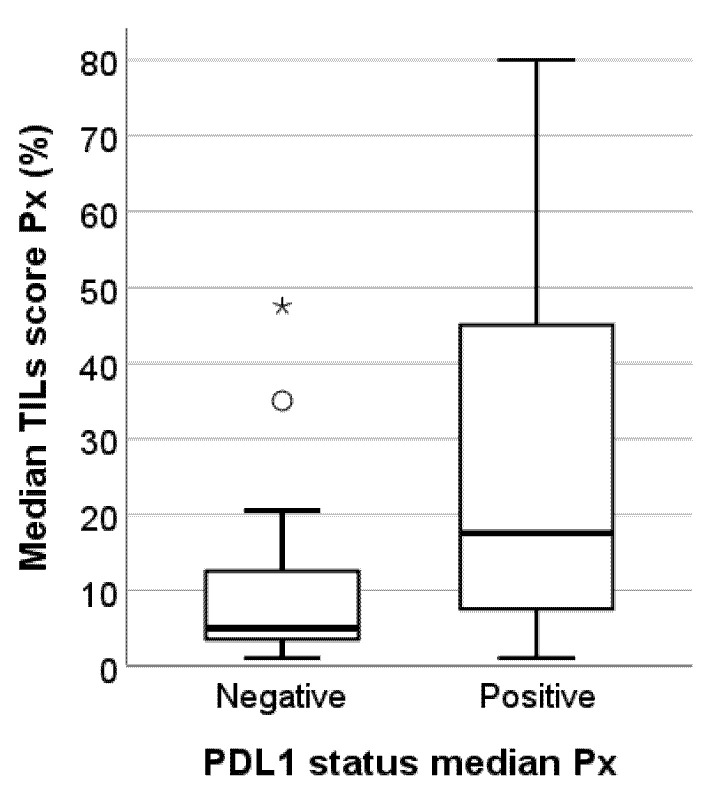
Box-and-whisker plot illustrating the median TILs scores Px for the PD-L1-negative and PD-L1-positive subgroups in this study cohort of 49 TNBC metastases. PD-L1-positive TNBC metastases have significantly higher TILs levels (*p* = 0.004). Circles represent outliers; asterisks represent extremes, * outliers, ° extremes.

**Figure 5 cancers-13-04910-f005:**
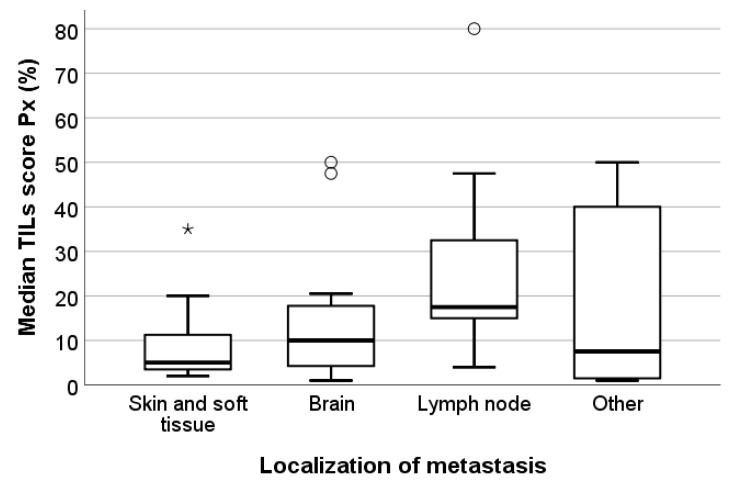
Box-and-whisker plot illustrating the median TILs scores Px per localization of the metastases in the study set. Although there is a trend towards higher TILs levels in lymph node metastases, there was no statistically significant association between both parameters (*p* = 0.136). Circles represent outliers; asterisks represent extremes. Bold lines within each box are the median P50. Skin and soft tissue metastases: *n* = 15. Brain metastases: *n* = 12. Lymph node metastases: *n* = 9. Metastases of other locations: *n* = 13 (including liver, bowel, kidney, and lung metastases), * outliers, ° extremes.

**Figure 6 cancers-13-04910-f006:**
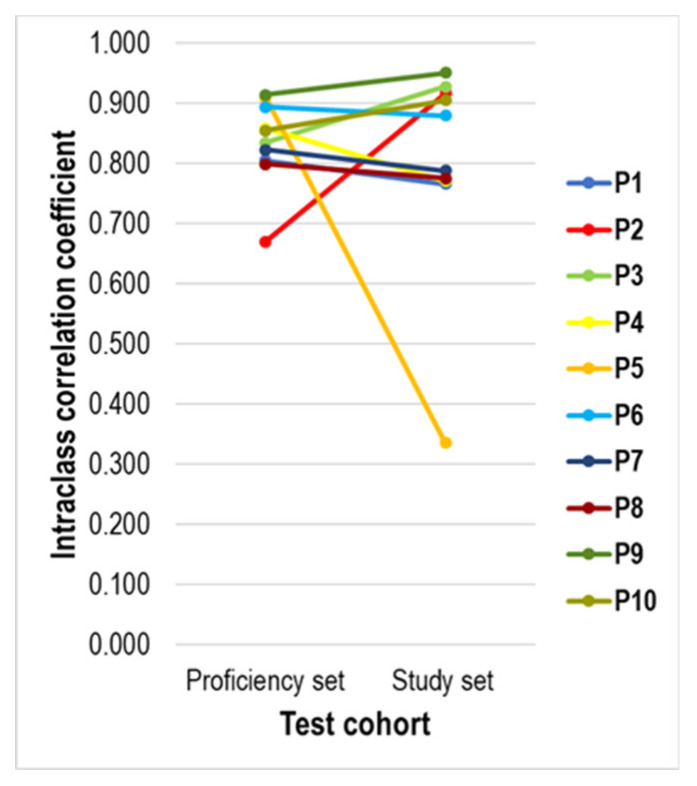
Line diagram showing the evolution between the agreement of the PD-L1 score of each pathologist with the median Px PD-L1 score in the proficiency set and the study set. The PD-L1 assessment of most pathologists showed a similar degree of agreement, except for P2 (higher agreement in the study set) and P5 (higher agreement in the proficiency set).

**Table 1 cancers-13-04910-t001:** Intraclass correlation coefficients for PD-L1/SP142 assessment as percentages in the proficiency set, as evaluated by ten participating pathologists.

PD-L1 Score (%)	PD-L1 P1 (%)	PD-L1 P2 (%)	PD-L1 P3 (%)	PD-L1 P4 (%)	PD-L1 P5 (%)	PD-L1 P6 (%)	PD-L1 P7 (%)	PD-L1 P8 (%)	PD-L1 P9 (%)	PD-L1 P10 (%)	PD-L1 Consensus (%)	Median PD-L1 Px (%)
PD-L1 P1 (%)	1.000	0.826	0.458	0.773	0.779	0.826	0.775	0.616	0.766	0.563	0.764	0.805
PD-L1 P2 (%)	0.826	1.000	0.387	0.547	0.631	0.736	0.631	0.386	0.594	0.510	0.623	0.670
PD-L1 P3 (%)	0.458	0.387	1.000	0.798	0.631	0.684	0.602	0.611	0.812	0.837	0.826	0.835
PD-L1 P4 (%)	0.773	0.547	0.798	1.000	0.770	0.743	0.716	0.693	0.828	0.705	0.800	0.858
PD-L1 P5 (%)	0.779	0.631	0.631	0.770	1.000	0.785	0.766	0.793	0.793	0.744	0.786	0.909
PD-L1 P6 (%)	0.826	0.736	0.684	0.743	0.785	1.000	0.826	0.557	0.761	0.653	0.811	0.894
PD-L1 P7 (%)	0.775	0.631	0.602	0.716	0.766	0.826	1.000	0.652	0.738	0.596	0.793	0.823
PD-L1 P8 (%)	0.616	0.386	0.611	0.693	0.793	0.557	0.652	1.000	0.677	0.611	0.756	0.799
PD-L1 P9 (%)	0.766	0.594	0.812	0.828	0.793	0.761	0.738	0.677	1.000	0.922	0.858	0.914
PD-L1 P10 (%)	0.563	0.510	0.837	0.705	0.744	0.653	0.596	0.611	0.922	1.000	0.772	0.855

P = pathologist.

**Table 2 cancers-13-04910-t002:** Intraclass correlation coefficients for PD-L1/SP142 assessment as percentages in the study set, as evaluated by ten participating pathologists. Because an ad hoc consensus score was lacking, the median Px PD-L1 score was used as a surrogate consensus score.

PD-L1 Score (%)	P1	P2	P3	P4	P5	P6	P7	P8	P9	P10	Median Px
P1	1.000	0.723	0.593	0.681	0.167	0.511	0.534	0.726	0.702	0.559	0.766
P2	0.723	1.000	0.766	0.706	0.432	0.693	0.713	0.809	0.868	0.746	0.917
P3	0.593	0.766	1.000	0.693	0.268	0.984	0.831	0.713	0.893	0.944	0.928
P4	0.681	0.706	0.693	1.000	0.438	0.662	0.715	0.649	0.699	0.661	0.771
P5	0.167	0.432	0.268	0.438	1.000	0.210	0.480	0.271	0.200	0.195	0.336
P6	0.511	0.693	0.984	0.662	0.210	1.000	0.823	0.676	0.848	0.938	0.880
P7	0.534	0.713	0.831	0.715	0.480	0.823	1.000	0.717	0.642	0.674	0.788
P8	0.726	0.809	0.713	0.649	0.271	0.676	0.717	1.000	0.703	0.624	0.775
P9	0.702	0.868	0.893	0.699	0.200	0.848	0.642	0.703	1.000	0.940	0.951
P10	0.559	0.746	0.944	0.661	0.195	0.938	0.674	0.624	0.940	1.000	0.905

P = pathologist.

**Table 3 cancers-13-04910-t003:** Cohen’s Kappa values per pathologist duo for PD-L1 status in the study set.

PD-L1 Status	P1	P2	P3	P4	P5	P6	P7	P8	P9	P10	Median Px
P1	1.000	0.620	0.762	0.704	0.559	0.776	0.500	0.535	0.952	0.507	0.866
P2	0.620	1.000	0.547	0.673	0.550	0.508	0.633	0.510	0.547	0.590	0.672
P3	0.762	0.547	1.000	0.630	0.481	0.604	0.514	0.383	0.712	0.692	0.780
P4	0.704	0.673	0.630	1.000	0.632	0.754	0.633	0.510	0.630	0.672	0.836
P5	0.559	0.550	0.481	0.632	1.000	0.455	0.593	0.386	0.481	0.623	0.707
P6	0.776	0.508	0.604	0.754	0.455	1.000	0.471	0.426	0.692	0.578	0.662
P7	0.500	0.633	0.514	0.633	0.593	0.471	1.000	0.633	0.514	0.553	0.634
P8	0.535	0.510	0.383	0.510	0.386	0.426	0.633	1.000	0.547	0.263	0.590
P9	0.952	0.547	0.712	0.630	0.481	0.692	0.514	0.547	1.000	0.429	0.780
P10	0.507	0.590	0.692	0.672	0.623	0.578	0.553	0.263	0.429	1.000	0.662

P = pathologist.

**Table 4 cancers-13-04910-t004:** Intraclass correlation coefficients for TILs assessment as percentages in the study set, as evaluated by ten participating pathologists. Because an ad hoc consensus score was lacking, the median Px TILs score was used as a surrogate consensus score.

TILs Assessment (%)	P1	P2	P3	P4	P5	P6	P7	P8	P9	P10	Median Px (%)
P1	1.000	0.867	0.527	0.474	0.583	0.859	0.804	0.444	0.612	0.663	0.851
P2	0.867	1.000	0.407	0.550	0.637	0.822	0.753	0.521	0.578	0.553	0.835
P3	0.527	0.407	1.000	0.232	0.393	0.397	0.556	0.295	0.533	0.555	0.618
P4	0.474	0.550	0.232	1.000	0.416	0.495	0.567	0.545	0.541	0.657	0.670
P5	0.583	0.637	0.393	0.416	1.000	0.609	0.606	0.553	0.338	0.527	0.753
P6	0.859	0.822	0.397	0.495	0.609	1.000	0.747	0.474	0.560	0.556	0.826
P7	0.804	0.753	0.556	0.567	0.606	0.747	1.000	0.491	0.682	0.694	0.908
P8	0.444	0.521	0.295	0.545	0.553	0.474	0.491	1.000	0.408	0.525	0.615
P9	0.612	0.578	0.533	0.541	0.338	0.560	0.682	0.408	1.000	0.482	0.756
P10	0.663	0.553	0.555	0.657	0.527	0.556	0.694	0.525	0.482	1.000	0.770

P = pathologist.

**Table 5 cancers-13-04910-t005:** Exploratory analysis of four different thresholds for TILs dichotomization and their influence on the concordance among all ten participating pathologists.

Threshold for TILs Dichotomization	Concordance Rate for All Pairs of Pathologists Sample Mean ± Standard Deviation
TILs <5 vs. ≥5%	0.734 (±0.042)
TILs <10 vs. ≥10%	0.733 (±0.062)
TILs <30 vs. ≥30%	0.797 (±0.067)
TILs <75 vs. ≥75%	0.925 (±0.053)

**Table 6 cancers-13-04910-t006:** Exploratory analysis of three different thresholds for dichotomization of PD-L1 scores, and their influence on the concordance among all ten participating pathologists.

Threshold for PD-L1 Dichotomization	Concordance Rate for All Pairs of Pathologists Sample Mean ± Standard Deviation	
	Proficiency Set	Study Set
PD-L1 <1 vs. ≥1%	0.961 (±0.030)	0.793 (±0.062)
PD-L1 <5 vs. ≥5%	0.804 (±0.084)	0.824 (±0.093)
PD-L1 <10 vs. ≥10%	0.814 (±0.067)	0.844 (±0.112)

**Table 7 cancers-13-04910-t007:** Kappa values for the ten participating pathologists in the exploratory analysis of the combined biomarker based on PD-L1 status (<1% vs. ≥1%) and TILs (<5% vs. ≥5%). Only those TNBC metastases with PD-L1/SP142 scores of ≥1% and ≥5% TILs were considered positive, and agreement was investigated.

Combined Biomarker	P1	P2	P3	P4	P5	P6	P7	P8	P9	P10
P1	1.000	0.613	0.852	0.718	0.511	0.719	0.355	0.548	0.791	0.641
P2	0.613	1.000	0.623	0.788	0.506	0.461	0.508	0.546	0.493	0.580
P3	0.852	0.623	1.000	0.724	0.435	0.632	0.371	0.468	0.733	0.745
P4	0.718	0.788	0.724	1.000	0.439	0.645	0.449	0.564	0.573	0.674
P5	0.511	0.506	0.435	0.439	1.000	0.448	0.671	0.371	0.468	0.563
P6	0.719	0.461	0.632	0.645	0.448	1.000	0.378	0.399	0.570	0.580
P7	0.355	0.508	0.371	0.449	0.671	0.378	1.000	0.546	0.493	0.580
P8	0.548	0.546	0.468	0.564	0.371	0.399	0.546	1.000	0.500	0.509
P9	0.791	0.493	0.733	0.573	0.468	0.570	0.493	0.500	1.000	0.780
P10	0.641	0.580	0.745	0.674	0.563	0.580	0.580	0.509	0.780	1.000

## Data Availability

Data are contained within the article and Appendix A. Additionally, a single file containing all raw data can be obtained upon reasonable request from the corresponding author.

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
