# Peer review of "Interobserver Agreement of PD-L1/SP142 Immunohistochemistry and Tumor-Infiltrating Lymphocytes (TILs) in Distant Metastases of Triple-Negative Breast Cancer: A Proof-of-Concept Study. A Report on Behalf of the International Immuno-Oncology Biomarker Working Group"

_cancers, 2021, doi:10.3390/cancers13194910_

Round 1
Reviewer 1 Report
The manuscript submitted by Van Bockstal et al. is an important and timely report on the inter-observational discordance among pathologists that may impact the decision-making in immunotherapy. The study is well designed and relevant to a larger audience of the journal.
The manuscript requires minor revision:
Patients with advanced TNBC benefits from atezolizumab treatment in those cases where the tumor contains ≥1% of PD-L1/SP142-positive immune cells. However, the treatment response to atezolimab depends on several other biomarkers, especially CD11b+ cells, as demonstrated in lung cancer (Bocanegra et al. IJMS 2019). This reviewer suggests adding information regarding other immune cells status in TNBC related to atezolizumab treatment response.
Authors should consider commenting on the degree of discordance between PD-L1 assessment vs. TILs assessment in the study cohort and their impact on the combined biomarker score. If any, it would be interesting to observe ICC for PD-L1/SP142 assessment and TILs vary among the pathologists similarly compared to median Px PD-L1 score and median Px TILs score, respectively.
There are many typos in the manuscript that needs to be corrected.
Reviewer 2 Report
This is a proof of concept study investigating the interobserver agreement in IHC assessment of PD-L1 using the SP142 assay and stroma TIL assessment via H&E assessment. The manuscript is well written and organized, and easy to understand even for a non-pathologists. The statistical methodology is sound and in concordance with other published interobserver analyses. The discussion is not over-inflated and is reflected by the results.
The problem is a very real one. In clinical trials, previously obtained archival tissue is frequently used for trial screening purposes, and this is typically most abundant and available from a primary tumour resection specimen. It could be argued that a more contemporary assessment can be obtained through a new metastatic biopsy – and thus, understanding the metastatic niche is of crucial importance – and one that should be considered when deciding to treat with immunotherapy based approaches based on a predictive biomarker.
Minor limitations:
- Small sample size and small number of pathologists. The authors however correctly identify that this is a proof of concept study and validation in larger datasets would be useful in the future
- It may be useful to compare these interobserver results with those of other PD-L1 IHC antibodies if this is available
Other:
Typo line 131: “Considering that the PD-L1/SP42….”
Line 164: Abbreviation “ER” – needs definition of this abbreviation
